# Bacterial Communities Found in Pit-Wall Mud and Factors Driving Their Evolution

**DOI:** 10.3390/foods12071419

**Published:** 2023-03-27

**Authors:** Hao Zhou, Boyang Xu, Shanshan Xu, Suwei Jiang, Dongdong Mu, Xuefeng Wu, Xingjiang Li

**Affiliations:** 1Anhui Fermented Food Engineering Research Center, School of Food and Biological Engineering, Hefei University of Technology, Hefei 230009, China; 2Department of Biotechnology and Food Engineering, Hefei University, Hefei 230601, China

**Keywords:** strong-flavor Baijiu, pit mud, bacteria, physicochemical property, maturation

## Abstract

Pit-wall mud (PWM) fosters bacterial communities involved in Baijiu production. PWM varies depending on pit age and height. In this study, we explored the bacterial communities in PWM and factors driving their evolution. The abundance and diversity of bacterial communities were low in new PWM (NPWM). In old PWM (OPWM), similar but diverse bacterial communities were observed at different heights. *Lactobacillus* was the predominant genus in NPWM, and *Caproiciproducens*, *Aminobacterium*, *Hydrogenispora*, *Lactobacillus*, *Petrimonas*, *Syntrophomonas*, and *Sedimentibacter* were the dominant genera in OPWM. A decrease was noted in the abundance of *Lactobacillus*, which indicated evolution. Among all the physicochemical properties, pH had the highest degree of interpretation with an R^2^ value of 0.965. pH also exerted the strongest effect on bacterial communities. The path coefficients of pH on bacterial community diversity and abundance were 0.886 and 0.810, respectively. *Caproiciproducens* and *Clostridium sensu stricto 12* metabolized lactic acid, inhibiting the growth of *Lactobacillus* at a suitable pH, which led to the maturation of PWM. Our findings enrich the literature on the evolution of bacterial communities in PM and the maturation of PM.

## 1. Introduction

Strong-flavor Baijiu (SFB), a major type of Chinese Baijiu, owes its flavor to its unique fermentation conditions [1]. SFB is obtained by distillation of fermented grains and the entire process of grain fermentation is conducted in a closed pit surrounded by pit mud (PM) [2]. PM fosters anaerobic bacteria involved in the fermentation of fermented grains [3,4]. The anaerobes in PM produce flavor compounds (e.g., caproate, butyrate, and 1-hexanol) [5], which contribute significantly to the quality of SFB. Therefore, the regulation of PM quality is an effective approach for improving the quality of SFB. Older pits [6,7] and deeper locations of pits [8] result in higher-quality SFB; this implies that the quality of PM varies depending on pit age and height.

The evolution of PM has been extensively studied in the past decade. Microbial communities evolve with aging PM over a period of 25 years [9]. Old PM exhibits higher microbial diversity than new PM and maintains a relatively stable state during long-term fermentation [10]. *Lactobacillus* is the predominant genus in new PM, whereas *Petrimonas*, clostridial cluster IV, *Sedimentibacter*, and *Methanoculleus* are the dominant genera in old PM [9]. According to location, PM at the bottom of a pit is commonly named pit-bottom mud (PBM), and PM on the four walls of a pit is pit-wall mud (PWM). The evolutionary patterns of the bacterial communities found in PBM have been analyzed in both vertical and horizontal dimensions. The maturation of new PBM starts at its corner first; *Caproiciproducens* is the predominant genus in the corners of both new and old PBM [11]. PWM exerts weaker effects on fermented grains than PBM; this is because of the lack of yellow water, which serves as medium [12], and because the large area of contact with fermented grains confer PWM with relatively strong potential for improving the quality of SFB. The evolutionary patterns of bacterial communities in the PWM of different ages were similar to those noted in PBM [13]; however, differences were noted between the PWM of different heights [14]. *Lactobacillus*, *Clostridium sensu stricto*, *Syntrophomonas*, and *Sedimentibacter* were the dominant genera in new PWM (NPWM) [15], whereas *Caproiciproducens*, unclassified Clostridiaceae 1, *Syntrophomonas*, *Proteiniphilum*, and *Petrimonas* were the dominant genera in old PWM (OPWM) [16].

To the best of our knowledge, the evolution of bacterial communities with the conversion of NPWM to OPWM has never been investigated by focusing simultaneously on the effects of pit age and height on their evolution. Changing the physicochemical properties of PM appears to be an effective approach for driving the evolution of bacterial communities in PM [16], which may facilitate the maturation of PM.

In this study, we explored the evolution of bacterial communities in PWM by focusing on the effects of pit age and height on their evolution. The evolution of these communities was driven by changing the key physicochemical properties of PM. This study enriches the literature on the evolution of bacterial communities in PM and the maturation of PM.

## 2. Materials and Methods

### 2.1. PWM Samples

PWM samples were collected from a well-known SFB manufacturer in Fuyang, Anhui Province, China. Three NPWM (3-year-old) and three OPWM (30-year-old) samples were randomly selected. The pit wall was divided into three layers, upper (U), middle (M), and lower (L), according to its height. The PWM samples were collected from the four walls of pits. Appendix A depicts the specific sampling points on each wall. Samples obtained from the same layer were pooled. Six types of PWM (N-U, N-M, N-L, O-U, O-M, and O-L) samples were evaluated in triplicate. The samples were rapidly packed into sterile anaerobic bags and stored at −80 °C for subsequent experiments.

### 2.2. Illumina MiSeq Sequencing

The total genomic DNA of the bacterial communities in PM was extracted using a FastDNA Spin kit. The extracted DNA was evaluated for purity and concentration. Next, polymerase chain reaction (PCR) was performed to amplify the V3–V4 variable regions of the 16S rRNA gene with the primers 338F_ (5′-ACTCCTACGGGAGGCAGCAG-3′) and 806R_ (5′-GGACTACHVGGGTWTCTAAT-3′). The amplification conditions were as follows: initial denaturation at 95 °C for 3 min, followed by 35 cycles of denaturation at 95 °C for 30 s, annealing at 55 °C for 30 s, extension at 72 °C for 45 s, and final extension at 72 °C for 10 min. After the reaction, the amplicons were maintained at 10 °C.

The PCR products maintained in Tris-HCl buffer were identified and purified through agarose gel electrophoresis (2%) and then quantified using QuantiFluor-ST considering the electrophoresis results. Samples with volumes suitable for sequencing were sequenced on the Illumina MiSeq PE300 platform.

### 2.3. Assessment of Physicochemical Properties

We selected eight primary physicochemical properties, namely moisture, pH, total acid, available phosphorus (AP), NH_4_^+^-N, humus, lactic acid, and acetic acid.

Moisture was measured by drying the PM samples at 100 °C for 3 h.

PM was soaked in deionized water (*w*/*v* ratio, 1:5) and sonicated for 10 min, the supernatant was used for pH and total acid detection. pH was measured by analyzing the supernatant using a pH meter. The supernatant was titrated with 0.1 mol/L NaOH using phenolphthalein as the indicator, then the total acid was calculated based on the consumption of NaOH.

AP in PM was extracted by soaking in NH_4_F–HCl solution, then the AP content was colorimetrically determined by measuring the absorbance of NH_4_F–HCl solution at a wavelength of 725 nm (DB34/T 2266-2014).

NH_4_^+^-N in PM was extracted by soaking in NaCl solution and the ammonia in the solution could react with K_2_HgI_4_ in a yellow color, then the NH_4_^+^-N content was colorimetrically determined by measuring the absorbance of the solution at a wavelength of 425 nm.

Humus in PM was extracted by soaking in Na_4_P_2_O_7_ solution, then the soluble sodium humate in the solution could be oxidized into CO_2_ and H_2_O by K_2_Cr_2_O_7_ in concentrated sulfuric acid. The humus content was calculated based on the consumption of K_2_Cr_2_O_7_ (DB34/T 2265-2014).

Lactic and acetic acids were identified using a high-performance liquid chromatography system equipped with an Ultimate LP-C18 column. PWM was mixed with 15% (*v*/*v*) CH_3_OH at a ratio of 1:9 and sonicated for 10 min. Afterward, it was filtered through a 0.22-μm organic filter membrane; the filtrate was used for further analyses. The mobile phase comprised CH_3_OH in potassium phosphate buffer at 1:19 (*v*/*v*). The chromatographic parameters were as follows: column temperature, 30 °C; flow rate, 0.6 mL/min; injection volume, 15 µL; limit of quantification, 5.0 µg/mL; limit of detection, 1.6µg/mL; and detection wavelength, 210 nm.

### 2.4. Statistical Analysis

Alpha-diversity indices were calculated using mothur (version 1.30.2). Principal coordinate analysis (PCoA) and the Wilcoxon rank-sum test were performed using R (version 4.2.0). The correlation network was visualized using Gephi (version 0.9). Redundancy analysis (RDA) was performed using Canoco (version 5.0). A heatmap was plotted using TBtools (version 0.665). All statistical analyses were performed using IBM SPSS Statistics (version 26).

Structural equation modeling (SEM) was performed using IBM SPSS Amos (version 24). The initial model assumed that all possible paths were reasonable; nonsignificant paths were gradually removed from the model until all of the remaining paths were significant. The model was continually revised according to the modification indices and estimates in the Amos output. The final model exhibited a low Akaike information criterion. Model fitness was measured using the chi-square test (*p* > 0.05), goodness-of-fit index (GFI > 0.90), comparative fit index (CFI > 0.95), Tucker–Lewis coefficient index (TLI > 0.90), and root mean square error of approximation (RMSEA < 0.05).

### 2.5. Driving the Evolution of Bacterial Communities in PWM

The composition of the medium used to drive the evolution of bacterial communities was as follows (per L): glucose, 5.0 g; lactic acid, 10.0 g; tryptone, 5.0 g; K_2_HPO_4_, 0.50 g; NH_4_Cl, 1.0 g; MgSO_4_, 0.20 g; NaCl, 0.50 g; FeSO_4_, 0.010 g; MnSO_4_, 0.010 g; CaCl_2_, 0.010 g; ZnSO_4_, 0.002 g; CoCl_2_, 0.002 g; CuSO_4_, 0.001 g; H_3_BO_3_, 0.001 g; and yellow water, 20 mL. The pH of the medium was adjusted to 6.0. Then, the medium was sterilized at 121 °C for 20 min. Three types of initial PWM samples were prepared for the experiment: (i) I-N, made by mixing N-U, N-M, and N-L at a 1:1:1 (*w*/*w*) ration; (ii) I-O, made by mixing O-U, O-M, and O-L at a 1:1:1 (*w*/*w*) ration; (iii) I-NO, made by mixing I-N and I-O at a 1:1 (*w*/*w*) ratio. The experiment was performed in triplicate. In brief, 4.0 g of initial PWM and 50 mL of medium were added to a 100-mL conical flask and anaerobically incubated for 14 days at 37 °C. Next, the evolved cultures (E-N, E-O, and E-NO) were centrifuged at 7000 rpm for 5 min. The supernatant was used to analyze organic acids, and the pellet was used to assess bacterial communities as previously reported in Section 2.2.

### 2.6. Accession Numbers

The raw sequence data were submitted to the Sequence Read Archive of the National Center for Biotechnology Information database under BioProject PRJNA907165 (BioSample accession numbers: SAMN31919453–SAMN31919461) and BioProject PRJNA912071 (BioSample accession numbers: SAMN32082417–SAMN32082434).

## 3. Results

### 3.1. Diversity of Bacterial Communities in Different PWM Samples

The minimum sample sequence number was 43,138 and the coverage curve of each sample reached a plateau (Appendix A); this indicated that the results of Illumina MiSeq sequencing were satisfactory.

The alpha-diversity of bacterial communities in different PWM samples was assessed using the ACE, Chao1, Shannon, and Simpson indices (Figure 1A). The bacterial communities present in NPWM varied significantly depending on pit height; their abundance and diversity decreased with decreasing height. By contrast, the bacterial communities present in OPWM did not vary significantly depending on pit height. The abundance and diversity of bacterial communities were higher in OPWM than in NPWM in the U, M, and L layers.

PCoA revealed the variability of bacterial communities among the PWM samples (Figure 1B). Significant differences were observed between NPWM and OPWM; however, PWM samples of the same age did not vary. PC1 (81.23%) exhibited a high degree of explanation for sample clustering; on the basis of PC1, NPWM and OPWM were clustered separately. On the basis of PC2 (10.94%), NPWM communities were clustered and OPWM communities were dispersed; no differences were noted between NPWM and OPWM.

### 3.2. Composition of Bacterial Communities in Different PWM Samples

Bacterial communities present in different PWM samples were similar at the phylum level (Appendix A). Firmicutes (99.26%) was the predominant phylum in NPWM, and Firmicutes (74.61%), Bacteroidota (12.95%), and Synergistota (9.79%) were the dominant phyla in OPWM.

The composition of bacterial communities varied significantly (at the genus level) between NPWM and OPWM (Figure 1C). The relative abundance of only three bacterial genera in NPWM was >1%, accounting for 96.40% of the overall bacterial communities. Only *Lactobacillus* was present in N-U, N-M and N-B, with a relative abundance of >1%. By contrast, two dominant genera, *Clostridium sensu stricto 12* and *Caproiciproducens*, were present in N-U, with relative abundances of >1%. *Lactobacillus* had the highest abundance in NPWM, and its relative abundance increased with decreasing pit height, with relative abundances of 79.84%, 96.09%, and 99.36% in N-U, N-M, and N-L, respectively. A total of 18 genera with relative abundances of >1% were found in OPWM, which together accounted for 78.42% of all bacterial genera. *Caproiciproducens*, *Aminobacterium*, *Hydrogenispora*, *Lactobacillus*, *Petrimonas*, *Syntrophomonas*, and *Sedimentibacter* were present in O-U, O-M, and O-L, with relative abundances of >1%. *Caproiciproducens* had the highest relative abundance in OPWM, with relative abundances of 9.11%, 25.87%, and 14.19% in O-U, O-M, and O-L, respectively.

### 3.3. Differences and Interactions between NPWM and OPWM

The diversity and composition of different PWM indicated that they varied primarily in terms of pit age rather than pit height (Figure 1). The Wilcoxon rank-sum test was performed to identify the genera with different abundances in NPWM and OPWM (Figure 2A). A total of 189 genera varied significantly; of them, 15 genera exhibited differences of >10%. *Lactobacillus* had a higher relative abundance in NPWM than in OPWM; this genus exhibited the highest significant difference. The remaining 14 genera had higher relative abundance in OPWM than in NPWM.

Correlation networking was performed to investigate the correlations between the selected 15 genera (Figure 2B). They were grouped according to the differences in their abundances between NPWM and OPWM. *Lactobacillus* was grouped under NPWM, whereas the remaining 14 genera were grouped under OPWM. A total of 104 correlations were identified among the 15 genera. Of these 104 correlations, 90 were positive and 14 were negative. All positive correlations were found within the group under OPWM, whereas all negative correlations were found between *Lactobacillus* and the group under OPWM. *Lactobacillus* was the marker genus for the difference between NPWM and OPWM.

### 3.4. Physicochemical Properties of Different PWM Samples

The physicochemical properties of PWM varied depending on pit age and height (Table 1). In NPWM, moisture and NH_4_^+^-N content showed no difference at different heights, whereas lactic and acetic acid contents decreased with increasing height. In OPWM, moisture, AP content, and NH_4_^+^-N content decreased with increasing height. In both NPWM and OPWM, pH increased with increasing height, whereas total acid and humus contents increased with decreasing height. pH and NH_4_^+^-N content were lower, but total acid, humus, and lactic acid contents were higher in NPWM than in OPWM in the U, M, and L layers.

### 3.5. Correlations between Bacterial Communities and Physicochemical Properties

RDA was performed to investigate the correlations between bacterial communities and physicochemical properties in different PWM samples (Figure 3A). RDA1 (89.15%) had a high degree of explanation for the clustering of PWM. NPWM and OPWM were clustered separately. NPWM was clustered relatively tightly; similar findings were obtained through the PCoA (Figure 1B). The heatmap revealed the correlations between physicochemical properties of PWM and the 15 genera selected on the basis of the Wilcoxon rank-sum test (Figure 3B). *Lactobacillus* was in a separate category, and the remaining 14 genera were clustered. Lactic acid, total acid, humus, and moisture were positively correlated with NPWM and *Lactobacillus*; lactic acid exhibited a relatively strong correlation. By contrast, pH, acetic acid, NH_4_^+^-N, and AP were positively correlated with OPWM and the remaining 14 genera (excluding *Lactobacillus*); pH and NH_4_^+^-N exhibited relatively strong correlations.

### 3.6. Complex Correlations Observed in PWM

SEM was performed to investigate the direct and indirect correlations between pit age, height, physicochemical properties, and bacterial communities (Figure 4A). Because acetic acid shared no significant path with pit age, height, or bacterial communities, it was excluded from the analysis to improve the rationality of the model. The main parameters used in SEM indicated that the final model was reasonable and the prediction results were close to the actual results (Appendix A). Pit age exerted the strongest effect on NH_4_^+^-N but no significant effect on moisture; it exerted positive effects on pH, AP, and NH_4_^+^-N. Height exerted significant effects on all physicochemical properties; it exerted negative effect on pH; its effect was the strongest on total acid but the weakest on NH_4_^+^-N. Only pH and AP exerted significant effects on the bacterial communities and could explain most of the variation; the R^2^ values were 0.972 and 0.655 for diversity and abundance, respectively. pH was the most influential physicochemical property and exerted positive effects on both abundance and diversity.

### 3.7. Evolution of Bacterial Communities in Different PWM Samples

To verify the results of SEM regarding the factors driving the evolution of bacterial communities and the driving effect of OPWM on the maturation of NPWM in actual production, we drove the evolution of bacterial communities in PWM. Different bacterial communities exhibited evolutionary patterns at the genus level (Figure 4B). No differences were observed between I-N and E-N. *Lactobacillus* and *Clostridium sensu stricto 12* were the dominant genera in both I-N and E-N. E-O and E-NO had similar community compositions. The dominant genera in I-O and I-NO, excluding *Caproiciproducens*, *Clostridium sensu stricto 12*, and *Lactobacillus*, were no longer dominant in E-O and E-NO. *Caproiciproducens* and *Clostridium sensu stricto 12* had higher relative abundances in E-O and E-NO than in I-O and I-NO; by contrast, *Lactobacillus* had higher relative abundance in E-NO than in I-NO but lower relative abundance in E-O than in I-O.

## 4. Discussion

PM is crucial for the quality of SFB. PWM strongly influences the quality of SFB because of its large contact area with the fermented grains. Improving the quality of PWM by changing the physicochemical properties may help improve the quality of SFB. In this study, we observed complex changes and interactions between the bacterial communities present in PWM and the physicochemical properties of different (based on pit age and height) PWM samples. The evolutionary patterns and possible maturation effects of PWM were explored.

The unique flavor of SFB results from the key flavor substances produced by the bacterial communities present in PM [17]. The abundance and diversity of bacterial communities in NPWM decreased with decreasing height but did not change in OPWM. At different heights, the abundance and diversity of bacterial communities were higher in OPWM than in NPWM (Figure 1A); these findings were similar to those noted in PBM [11]. *Lactobacillus* was the predominant genus in NPWM at different heights, and its abundance increased with decreasing height (Figure 1C); this trend is inconsistent with that of another study [15]. *Lactobacillus* is the predominant genus in fermented grains and yellow water and can proliferate in anaerobic environments with low pH [18]. The invasion of *Lactobacillus* from fermented grains and yellow water might have resulted in its high abundance in NPWM because of insufficient lactic acid metabolism [19,20]. The deposition of yellow water resulted in a reduced pH and improved anaerobic environment in the L layer of PWM [9], which led to a higher abundance of *Lactobacillus*. *Caproiciproducens*, *Aminobacterium*, *Hydrogenispora*, *Lactobacillus*, *Petrimonas*, *Syntrophomonas*, and *Sedimentibacter* were the dominant genera in OPWM at different heights (Figure 1C), and similar community composition was observed in old PBM [13,21]. The abundance of bacterial genera capable of using lactic acid as a carbon source for growth and metabolism (e.g., *Caproiciproducens*) increased during the evolution from NPWM to OPWM [16]. Although the consumption of lactic acid by these bacteria increased the pH of PWM, other bacteria grew and multiplied [22], which explains the correlations in the correlation network (Figure 2B). *Lactobacillus* showed the most significant difference in abundance between NPWM and OPWM (Figure 2A). The changes in its relative abundance may be used as a marker for the evolution of PWM.

The changes in physicochemical properties reflect the changes in bacterial communities [23,24]. Different (in terms of pit age and height) PWM samples varied primarily in terms of pH, total acid, NH_4_^+^-N, and humus; similar findings were obtained when assessing PBM [9,11,25]. O-U and O-M were close to the top of the pit and, therefore, easily exposed to air; these layers lost moisture easily during the long-term fermentation process [14]. The deposition of water from fermented grains due to gravity increased the moisture of O-L (Table 1). pH was negatively correlated with height in both NPWM and OPWM (Table 1); this finding was inconsistent with that of another study [14]. The variation in the pH of PWM was strongly correlated with the lactic acid metabolism capacity of the dominant bacterial genera (Figure 3B). *Lactobacillus* produces large quantities of lactic acid, which has a lower dissociation constant and higher acid effect than enolic acids, such as acetic and butyric acids; therefore, an increase in lactic acid content results in a reduced pH [26]. NH_4_^+^-N serves as a nitrogen source for microbial growth and affects the community composition and metabolic function of the bacteria present in PM [27,28]. In OPWM, NH_4_^+^-N can be produced by highly abundant bacteria [29,30], such as *Sedimentibacter* and *Aminobacterium* (Figure 1C). Humus is an organic material formed through the decomposition of dead organisms, which are constantly decomposed by PWM microorganisms driven by the lack of supplementary sources [13]; thus, humus content decreases with increasing pit age (Table 1).

Complex interactions between microbial communities and physicochemical properties determine the quality of PM [31]. pH was found to be the most influential physicochemical property (Figure 4A). A suitable pH and carbon source can effectively change the composition of the bacterial communities present in PM and increase the production of ethyl caproate [32,33]. In this study, the relative abundance of *Lactobacillus* in I-N and E-N was >90% (Figure 4B). The species of *Lactobacillus* predominant in SFB production was *L. acetotolerans*, which promotes the production of ethyl acetate and ethyl lactate and metabolizes most of the glucose present in a system [34]. The bacteria capable of utilizing lactic acid exhibited low initial abundance and slow growth, thus leading to a consistently high abundance of *L. acetotolerans*. The relative abundance of *Lactobacillus* was higher in E-O than in I-O (Figure 4B) because of the low activity of *L. acetotolerans* [35], the rapid growth of *L. acidipiscis* using glucose [36], and *L. buchneri* using lactic acid [37] at neutral pH (Appendix A). Lactic acid was not detected in E-O or E-NO, which indicates that lactic acid was a suitable carbon source for the growth and metabolism of the bacteria present in OPWM. *Caproiciproducens* can metabolize lactic acid to caproic acid [22,32], and *Clostridium sensu stricto 12* can metabolize lactic acid to butyric acid [38]. Caproic acid, butyric acid, and neutral pH inhibit the growth and metabolism of *Lactobacillus* [39], which leads to the maturation of the bacterial communities in the mixed system containing both NPWM and OPWM.

We explored the pattern of the evolution of NPWM to OPWM in terms of pit age and height and identified the relevant driving factors and maturation effects. However, our study has some limitations. This was a small-scale study conducted in a laboratory setting. We could not analyze the quality of SFB produced by the evolved PWM culture because doing so is difficult in a laboratory setting. Furthermore, we could not assess the subsequent changes in the bacterial communities present in the evolved PWM and the physicochemical properties of the evolved PWM during long-term fermentation. Future studies should be conducted to overcome the aforementioned limitations. Furthermore, our findings should be extrapolated to an actual production setting; however, for this, considerable efforts in terms of feasibility assessment and condition simplification are necessary.

## 5. Conclusions

The PWM bacterial communities involved in the production of SFB exhibited unique evolutionary patterns depending on pit age and height. The abundance and diversity of bacterial communities decreased with decreasing height in NPWM but not in OPWM. OPWM had highly abundant and diverse bacterial communities compared to those in NPWM in the U, M, and L layers. *Lactobacillus* was the predominant genus in NPWM, and *Caproiciproducens*, *Aminobacterium*, *Hydrogenispora*, *Lactobacillus*, *Petrimonas*, *Syntrophomonas*, and *Sedimentibacter* were the dominant genera in OPWM. The decrease in the abundance of *Lactobacillus* indicated evolution. Moisture, lactic acid, total acid, and humus were positively correlated with the dominant bacterial genera in NPWM; in contrast, pH, acetic acid, NH_4_^+^-N, and AP were positively correlated with the dominant bacterial genera in OPWM. pH had the strongest effect on the bacterial communities found in PWM and increased with increasing pit age and height. *Caproiciproducens* and *Clostridium sensu stricto 12* increased in abundance after evolution and inhibited the growth of *Lactobacillus*, thus leading to the maturation of the bacterial communities in the mixed system containing both NPWM and OPWM. This study enriches the literature on the evolution of bacterial communities in PM and the maturation of PM.

## Figures and Tables

**Figure 1 foods-12-01419-f001:**
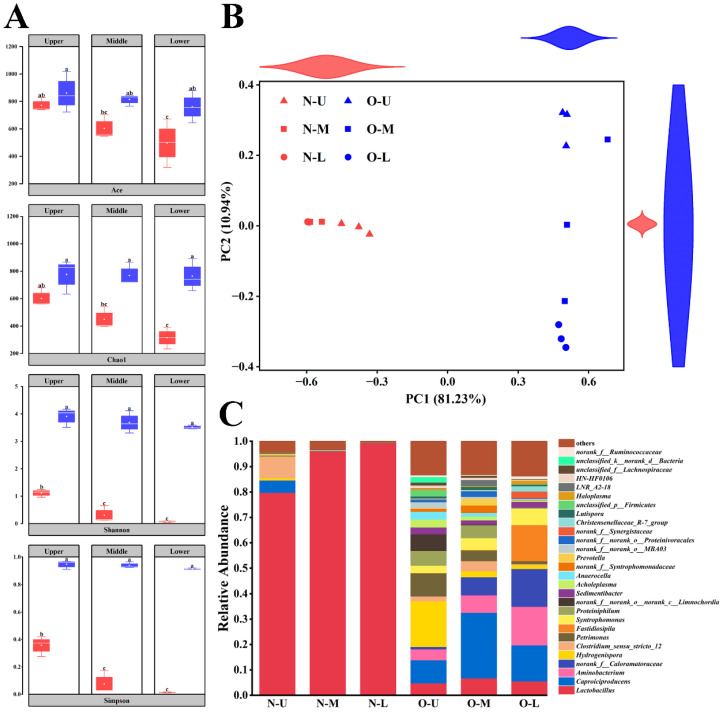
(**A**) Alpha-diversity of the bacterial communities (at the level of operational taxonomic unit [OTU]) present in new (red) and old (blue) pit-wall mud (PWM) samples. Considerable differences among various PWM samples are denoted by letters (abc) above the error bar. (**B**) Principal coordinate analysis (at the OTU level) performed using the weighted_unifrac distance matrix (R = 0.780; *p* = 0.001). (**C**) Composition of the bacterial communities (at the genus level) found in various PWM samples. N and O indicate new (3-year-old) and old (30-year-old) PWM samples, respectively. U, M, and L indicate the upper, middle, and lower layers of PWM, respectively.

**Figure 2 foods-12-01419-f002:**
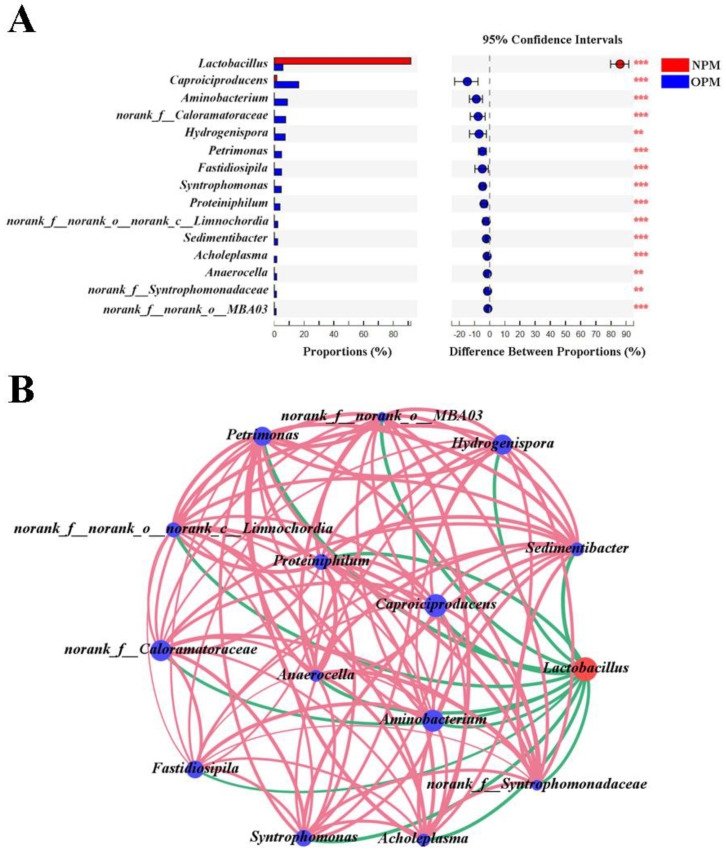
(**A**) Wilcoxon rank-sum test performed on the basis of a two-tailed test and false discovery rate correction revealed 15 bacterial genera with different abundances (difference > 10%) in new (red) and old (blue) pit-wall mud samples. Significance levels are denoted as follows: ** *p* < 0.01 and *** *p* < 0.001. (**B**) Correlation networks observed between the 15 bacterial genera (Spearman correlation coefficient [r] > 0.5; *p* < 0.05). The red line indicates a positive correlation, whereas the green line indicates a negative correlation. The thickness of the lines indicates the values of the correlation coefficients.

**Figure 3 foods-12-01419-f003:**
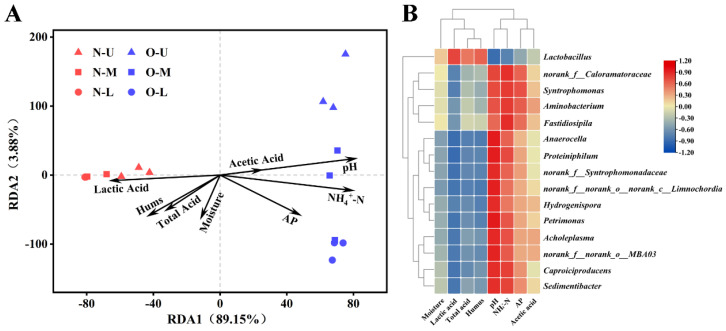
(**A**) Redundancy analysis revealed a correlation between the bacterial communities found in pit mud (PM) and the physicochemical properties of PM. N and O indicate new (3-year-old) and old (30-year-old) pit-well mud samples, respectively. U, M, and L indicate the upper, middle, and lower layers of PWM, respectively. (**B**) Heatmap constructed using Spearman correlation coefficients illustrates the correlations between physicochemical properties and the 15 bacterial genera selected on the basis of the Wilcoxon rank-sum test; the clustering of physicochemical properties and bacterial genera on the basis of complete indices.

**Figure 4 foods-12-01419-f004:**
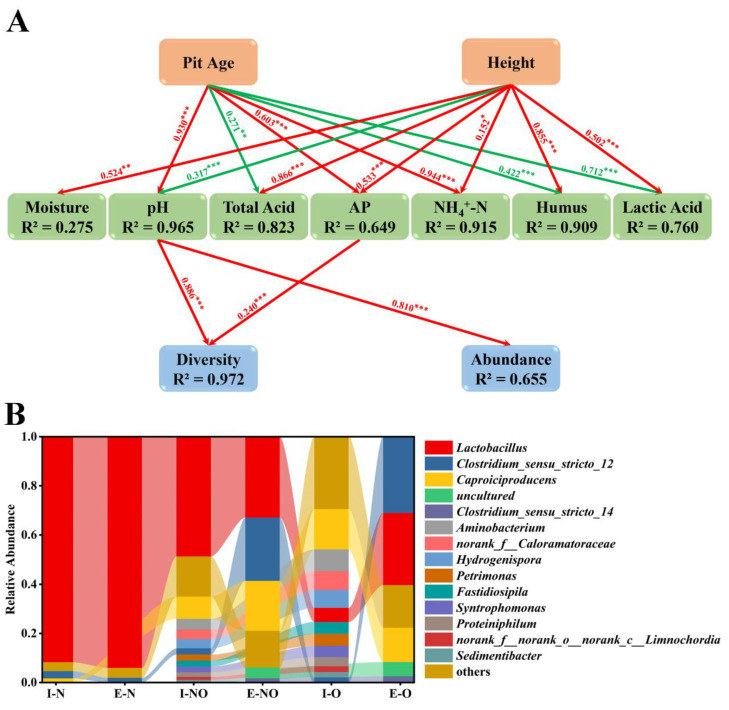
(**A**) Structural equation modeling revealed correlations between pit age, height, physicochemical properties, and bacterial communities. The red arrow indicates a positive correlation, whereas the green arrow indicates a negative correlation. Significance levels are denoted as follows: * *p* < 0.05, ** *p* < 0.01, and *** *p* < 0.001. The R^2^ values indicate the proportion of variance. (**B**) Values presented above the arrows indicate standardized path coefficients. Composition of the bacterial communities (at the genus level) found in different initial PWM and evolved PWM cultures. I and E indicate initial and evolved PWM cultures, respectively. N, O, and NO indicate NPWM, OPWM, and NPWM mixed with OPWM at a 1:1 (*w*/*w*) ratio, respectively.

**Table 1 foods-12-01419-t001:** Physicochemical properties of PWM samples. The superscript letters (a–f) on the upper-right side of the values mean the significant differences between PWM samples.

	Sample	NPWM			OPWM		
Parameters		Upper	Middle	Lower	Upper	Middle	Lower
Moisture (%)		39.78 ± 0.88 ^b^	39.81 ± 2.18 ^b^	39.52 ± 0.25 ^b^	35.94 ± 1.80 ^c^	36.29 ± 0.77 ^c^	44.05 ± 0.98 ^a^
pH		4.03 ± 0.05 ^d^	3.92 ± 0.05 ^e^	3.77 ± 0.00 ^f^	5.20 ± 0.00 ^a^	5.05 ± 0.06 ^b^	4.61 ± 0.06 ^c^
Total Acid (mg/g)		10.10 ± 0.37 ^c^	14.60 ± 0.28 ^b^	20.50 ± 0.57 ^a^	7.50 ± 0.13 ^d^	7.92 ± 0.91 ^d^	19.76 ± 0.51 ^a^
AP (mg/100 g)		241.07 ± 36.92 ^bc^	70.32 ± 9.74 ^d^	256.17 ± 58.25 ^bc^	196.37 ± 40.06 ^cd^	381.10 ± 62.01 ^b^	985.00 ± 113.87 ^a^
NH_4_^+^-N (mg/100 g)		26.97 ± 2.88 ^c^	35.09 ± 6.41 ^c^	24.82 ± 1.56 ^c^	61.03 ± 2.36 ^b^	75.01 ± 1.44 ^a^	80.48 ± 8.35 ^a^
Humus (%)		10.99 ± 0.85 ^b^	15.90 ± 0.55 ^a^	17.28 ± 1.29 ^a^	8.35 ± 0.66 ^c^	11.81 ± 0.58 ^b^	15.75 ± 0.60 ^a^
Lactic Acid (mg/g)		9.43 ± 0.29 ^c^	11.65 ± 2.30 ^b^	20.56 ± 0.50 ^a^	6.44 ± 0.18 ^d^	2.93 ± 0.60 ^e^	8.68 ± 0.33 ^c^
Acetic Acid (mg/g)		1.60 ± 0.51 ^b^	1.74 ± 0.08 ^b^	4.40 ± 0.33 ^a^	4.43 ± 0.42 ^a^	1.71 ± 0.27 ^b^	4.57 ± 0.95 ^a^

## Data Availability

Data is contained in the article or Appendix A.

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
