# Peer review of "Bacterial Communities Found in Pit-Wall Mud and Factors Driving Their Evolution"

_foods, 2023, doi:10.3390/foods12071419_

Round 1
Reviewer 1 Report
The manuscript is interesting and informative. The author described the isolation of microbes from pit wall mud. However the author needs explain initially about pit wall mud,pit bottom mud and the food involved in the mud .in the introduction itself the author should explain more then only the readers will know about the authors study.
Author Response
Dear reviewer,
Thank you for your review comments. Based on your review comments, we have made the following revised portions to the manuscript.
Point 1: The manuscript is interesting and informative. The author described the isolation of microbes from pit wall mud. However the author needs explain initially about pit wall mud,pit bottom mud and the food involved in the mud .in the introduction itself the author should explain more then only the readers will know about the authors study.
Does the introduction provide sufficient background and include all relevant references? Must be improved
Response 1: Based on your comments, we found that the Introduction to the research object in the manuscript was not accurate enough. We have revised the Introduction and emphasized the role of pit mud in the fermentation of Strong-flavor Baijiu in the first paragraph of the Introduction. In the second paragraph of the Introduction, we have described the pit bottom mud and pit wall mud in detail.
Point 2: Are the methods adequately described? Must be improved
Response 2: We have described our experimental method in more detail, especially the paragraph 2.3.

Reviewer 2 Report
General review:
The manuscript foods-2275167 “Bacterial Communities Found in Pit-Wall Mud and Factors Driving Their Evolution” has an "innovative objective". The manuscript presents modern and advanced molecular biology techniques (Illumina MiSeq sequencing), in addition to presenting well-elaborated results and discussion.
The present manuscript needs "minor revision".
Abstract: Bacterial genera and species are in italics. Please correct this. Check this in the manuscript.
Please add physical-chemical quantification in the abstract.
Author Response
Dear reviewer,
Thank you for your review comments. Based on your review comments, we have made the following revised portions to the manuscript.
Point 1: Bacterial genera and species are in italics. Please correct this. Check this in the manuscript.
Response 1: We have revised the bacterial genera and species involved in this article to italics.
Point 2: Please add physical-chemical quantification in the abstract.
Response 2: We have added a quantitative description of physicochemical properties in the abstract. Our focus is not on describing how the physicochemical properties change, but rather on how the physicochemical properties affect the bacterial communities in the pit mud. Therefore, we specifically introduce the impact of pH based on it is the most important physicochemical properties identified in our research.

Reviewer 3 Report
I evaluated the manuscript, which entitled “Bacterial Communities Found in Pit-Wall Mud and Factors Driving Their Evolution”.
The work is well written, consistent and with a very clear focus.
In the current form, need a minor revision the manuscript.
I completed the revision reporting a brief list of remarks.
Please, add the Author List and Affiliations as in the Instruction for Authors of Foods. Authors' full first and last names must be provided. The initials of any middle names can be added. The PubMed/MEDLINE standard format is used for affiliations: complete address information including city, zip code, state/province, and country. At least one author should be designated as the corresponding author. The email addresses of all authors will be displayed on published papers.
Please, provide the Highlights only for the Editor during submission.
Please keep attention to the manuscript typing (examples: L4, L31, L45-46, L 240, L306, L393).
L101-116: Please, provide more information about the methods also adding literature references.
L116: Authors should add the limit of detection (LOD) and the limit of quantification (LOQ) values.
L118: Mothur v1.30.2
L137-138: Please, explain more clearly if the sampling of NPWM and OPWM constituting the I-N and I-O were collected by mixing the Upper, Medium and Lower part of PWM to obtain homogenous samples.
L143: …to assess bacterial communities as previously reported in paragraph 2.2.
L301-305: I suggest to reformulate in a more clear way to simply verify the correspondence with data.
L346: “O-L” instead of “O-B”.
L370: I suggest to use italic for the species of bacteria (example L367 L. acetotolerans or L373 L. acidipiscis…) through the text.
Please, add the Author Contributions, Data Availability Statement, Conflicts of Interest as in the Instructions for Authors of Foods.
Moreover, Reference list should be reported as in the Instructions for Authors of Foods. In the text, reference numbers should be placed in square brackets [ ], and placed before the punctuation; for example [1], [1–3] or [1,3]. For embedded citations in the text with pagination, use both parentheses and brackets to indicate the reference number and page numbers; for example [5] (p. 10). or [6] (pp. 101–105). The reference list should include the full title and all authors. See the Reference List and Citations Guide for more detailed information.
In all other respects this is a fine work.
Author Response
Dear reviewer,
Thank you for your review comments. Based on your review comments, we have made the following revised portions to the manuscript.
Point 1: Please, add the Author List and Affiliations as in the Instruction for Authors of Foods. Authors' full first and last names must be provided. The initials of any middle names can be added. The PubMed/MEDLINE standard format is used for affiliations: complete address information including city, zip code, state/province, and country. At least one author should be designated as the corresponding author. The email addresses of all authors will be displayed on published papers.
Response 1: We have now added all the Author List and Affiliations in the manuscript.
Point 2: Please, provide the Highlights only for the Editor during submission.
Response 2: The Highlights have been removed from the manuscript.
Point 3: L101-116: Please, provide more information about the methods also adding literature references..
Response 3: We have now described the experimental method in detail. However, the methods we used are based on Chinese textbooks or local testing standards in Anhui Province, China, and there is no references can be added.
Point 4: L116: Authors should add the limit of detection (LOD) and the limit of quantification (LOQ) values.
Response 4: We have now added the limit of detection (LOD) and the limit of quantification (LOQ) values in the manuscript.
Point 5: L118: Mothur v1.30.2.
Response 5: Similar errors involved in the methods have been corrected.
Point 6: L118: L137-138: Please, explain more clearly if the sampling of NPWM and OPWM constituting the I-N and I-O were collected by mixing the Upper, Medium and Lower part of PWM to obtain homogenous samples..
Response 6: We have added a detailed description of I-N and I-O preparation.
Point 7: L143: …to assess bacterial communities as previously reported in paragraph 2.2.
Response 7: This has been modified in the manuscript.
Point 8: L301-305: I suggest to reformulate in a more clear way to simply verify the correspondence with data..
Response 8: The explanation in the manuscript has been revised to make it simpler and clearer.
Point 9: L346: “O-L” instead of “O-B”.
Response 9: This has been modified in the manuscript.
Point 10: L370: I suggest to use italic for the species of bacteria (example L367 L. acetotolerans or L373 L. acidipiscis…) through the text..
Response 10: Similar errors involved in the manuscript have been corrected..
Point 11: Please, add the Author Contributions, Data Availability Statement, Conflicts of Interest as in the Instructions for Authors of Foods.
Response 11: The Author Contributions, Data Availability Statement and Conflicts of Interest have been added to the manuscript according to the Instructions for Authors of Foods.
Point 12: Moreover, Reference list should be reported as in the Instructions for Authors of Foods. In the text, reference numbers should be placed in square brackets [ ], and placed before the punctuation; for example [1], [1–3] or [1,3]. For embedded citations in the text with pagination, use both parentheses and brackets to indicate the reference number and page numbers; for example [5] (p. 10). or [6] (pp. 101–105). The reference list should include the full title and all authors. See the Reference List and Citations Guide for more detailed information.
Response 12: All the references have been modified for the correct format as in the Instructions for Authors of Foods.
